# Estimation of Additional Costs in Patients with Ventilator-Associated Pneumonia

**DOI:** 10.3390/antibiotics13010002

**Published:** 2023-12-19

**Authors:** Ginger G. Cabrera-Tejada, Pablo Chico-Sánchez, Paula Gras-Valentí, Francisco A. Jaime-Sánchez, Maria Galiana-Ivars, Sonia Balboa-Esteve, Isel L. Gómez-Sotero, José Sánchez-Payá, Elena Ronda-Pérez

**Affiliations:** 1Preventive Medicine Service, Dr. Balmis General University Hospital, 03010 Alicante, Spain; 2Epidemiology Unit, Preventive Medicine Service, Dr. Balmis General University Hospital, Alicante Institute for Health and Biomedical Research (ISABIAL), 03010 Alicante, Spain; chico_pab@gva.es (P.C.-S.); gras_pau@gva.es (P.G.-V.); gomez_ise@gva.es (I.L.G.-S.); sanchez_jos@gva.es (J.S.-P.); 3Medical Intensive Care Unit, Dr. Balmis General University Hospital, Alicante Institute for Health and Biomedical Research (ISABIAL), 03010 Alicante, Spain; franciscoangelj@yahoo.es (F.A.J.-S.);; 4Anesthesiology Service and Surgical Intensive Care Unit, Dr. Balmis General University Hospital, Alicante Institute for Health and Biomedical Research (ISABIAL), 03010 Alicante, Spain; mgivars@gmail.com; 5Public Health Research Group, University of Alicante, San Vicente de Raspeig, 03690 Alicante, Spain; 6Biomedical Research Networking Center (CIBERESP), 28029 Madrid, Spain

**Keywords:** healthcare costs, pneumonia, ventilator associated, healthcare-associated infections, nosocomial infection

## Abstract

Healthcare-associated infections (HAIs) present a global public health challenge, contributing to high morbidity and mortality and substantial economic burdens. Ventilator-associated pneumonia (VAP) ranks as the second most prevalent HAI in intensive care units (ICUs), emphasizing the need for economic analyses in this context. This retrospective cohort study, conducted at the General Hospital of Alicante from 2012 to 2019, aimed to assess additional costs related to VAP by comparing the extended length of stay for infected and non-infected ICU patients undergoing mechanical ventilation (MV) for more than 48 h. Employing propensity score association, 434 VAP patients were compared to an equal number without VAP. The findings indicate a significantly longer mechanical ventilation period for VAP patients (17.40 vs. 8.93 days, *p* < 0.001), resulting in an extra 13.56 days of stay and an additional cost of EUR 20,965.28 per VAP episode. The study estimated a total cost of EUR 12,348,965.28 for VAP during the study period, underscoring the economic impact of VAP. These findings underscore the urgent need for rigorous infection surveillance, prevention, and control measures to enhance healthcare quality and reduce overall expenditures.

## 1. Introduction

Healthcare-associated infections (HAIs) represent a significant global public health challenge, contributing not only to significant morbidity and mortality but also to the substantial economic impact they generate [1]. These infections are characterized by being absent at the time of hospital admission and developing at least 48 h after admission to a healthcare facility [2,3,4]. According to the Spanish Prevalence of Nosocomial Infections Study (EPINE), the prevalence of HAIs in 2019 was 7.77% [5], data very similar to those reported by the World Health Organization (WHO), with figures ranging from 5 to 10% [6,7]. In Spain, it is estimated that around 4.5 million HAIs occur each year, resulting in 37,000 deaths, 16 million additional hospital stays, and an extra cost of EUR 700 million [8,9].

In critical care units (CCUs), including intensive care units (ICUs) and resuscitation units (RUs), the most prevalent HAIs are those associated with medical devices [4,5,10,11]. Of these, ventilator-associated pneumonia (VAP) stands as the second most prevalent, registering a frequency of 29.99% in 2019, only surpassed by central venous catheter-associated bloodstream infections (CVC-BSIs). Paradoxically, despite ranking second in terms of prevalence, VAP commands the highest attributed mortality rate (13%) among device-associated infections [12,13,14,15]. Several studies estimate that 5–25% of patients on mechanical ventilation develop VAP [15,16].

Beyond its substantial economic impact, VAP imposes a significant social burden on patients, their families, and healthcare systems as a whole [17]. The increased costs resulting from the development of VAP are primarily associated with prolonged ventilatory support requirements, extended stays in CCUs, and overall extended hospitalization. Leistner et al. (2013) reported an average length of stay for VAP cases of 36 days, translating to a 9-day extension of hospital stay and associated costs amounting to EUR 17,015 [18]. Additionally, Cocanour et al. (2005) reported that a single episode of VAP leads to expenses amounting to USD 57,000 [19].

The measurement of the economic impact of HAIs varies, but within the healthcare context, assessing the extension of hospital stays is considered a suitable parameter [10,20]. Obtaining reliable information regarding the variability and magnitude of this social and economic burden is crucial for enhancing decision-making in healthcare policies. Therefore, the primary focus of healthcare facilities and systems should be the surveillance, prevention, and control of HAIs through the implementation of prevention and care guidelines and protocols [21,22,23].

Despite finding and reviewing various studies conducted worldwide on the additional cost estimation, there are not many studies that specifically address the additional costs that HAIs, or more specifically VAP, directly impose on healthcare systems. In Spain, to date, we have not come across any studies that analyze the increase in hospital stay and the additional cost generated by the development of VAP in the Spanish healthcare system, which is considered one of the best healthcare systems worldwide [24,25]. Furthermore, on an international level, we have not found many studies that allow for the comparison of groups of patients that were as homogeneous as possible, as allowed by the propensity score (PS), which attributes the extra hospital stay to the development of VAP.

Conducting studies of this nature is essential, as it provides critical insights into the costs incurred by healthcare systems. Such research plays a pivotal role as a comparative foundation, as it enables examination at both the national and global levels and fosters a deeper understanding of healthcare systems worldwide, with the acknowledgment that not all healthcare systems have the same characteristics: Depending on the structure of the healthcare system, the cost implications may vary significantly. Furthermore, these studies serve to highlight weaknesses or shortcomings in different healthcare systems, requiring acknowledgment and targeted interventions to ensure continuous improvements in quality.

The objective of this study is to analyze the additional costs associated with the acquisition of VAP by calculating the extra length of stay for patients in a tertiary-level hospital.

## 2. Materials and Methods

A retrospective cohort study was conducted using data collected from the database of the ventilator-associated pneumonia (VAP) surveillance program, included in the healthcare-associated infection (HAI) surveillance system of the General University Hospital of Alicante (HGUA), the second-largest hospital in the Valencian Community, with 825 beds, providing healthcare services to a population of 274,271 residents and serving as a referral hospital for approximately 2,000,000 residents.

All patients admitted to the critical care units (CCUs) who underwent invasive mechanical ventilation (MV) for a period exceeding 48 h between the years 2012 and 2019 were included in the study. The definition of ventilator-associated pneumonia (VAP) followed the criteria established by the European Centre for Disease Prevention and Control (ECDC), which included clinical, radiological, and laboratory criteria [26]. An exposed patient was defined as any patient admitted to the CCU with invasive MV who developed VAP. A non-exposed patient was defined as any patient admitted to the CCU with invasive MV who did not develop VAP.

In order to address potential confounding variables and create a balanced comparison between patients with and without VAP, we employed a rigorous approach to calculate the propensity score (PS). The PS reflects the individual probability of developing VAP and was derived through a multistep process. Firstly, an association analysis was conducted using the Chi-square test to identify variables significantly associated with the development of VAP. These variables were drawn from a comprehensive set of risk factors established in the literature, including sex (male/female), age group (≥65 years/<65 years), and comorbidities such as diabetes mellitus (DM), chronic obstructive pulmonary disease (COPD), neoplasia, neutropenia (<1000 n/mm^3^), acute respiratory distress syndrome (ARDS), renal failure, multiorgan failure (MOF), history of tobacco use, history of alcoholism, history of trauma, immunodeficiency (primary or secondary, neutropenia <500 n/mm^3^, etc.—for a complete definition, refer to the EPINE 2019 protocol [27]), acute coronary syndrome (ACS), central venous catheter-associated bloodstream infection (CVC-BSI), exitus, and severity patient condition per Acute Physiology and Chronic Health Evaluation (APACHE) II score. For the association analysis, the presence of VAP (yes/no) was considered the outcome variable, and the risk factors associated with the development of VAP were considered the explanatory variables. The magnitude of the association was quantified using the odds ratio (OR) with a 95% confidence interval (CI).

Subsequently, variables that demonstrated statistical significance in this univariate analysis were subjected to further scrutiny in a multivariate logistic regression model, calculating the adjusted odds ratio (aOR) with a 95% CI. The final selection of variables included in the model was based on their sustained significance, ensuring a focus on factors with a significant impact on VAP development. The resulting variables that were statistically significant in the multivariate model were then used to calculate the propensity score for each patient. This score represents the likelihood of a patient developing VAP based on their individual risk factors. With the propensity score in hand, for every VAP patient, a non-exposed counterpart with a comparable PS was selected that was either associated with the exact PS score or had a difference of less than 0.03. The homogeneity of the resulting associated cohort was rigorously verified using the Chi-square test, providing confidence in the comparability of the VAP and non-VAP groups after matching based on the propensity score. The level of statistical significance used for hypothesis testing was set at *p* < 0.05. It is crucial to note that cases of VAP without a suitable counterpart, as determined by the PS, were excluded from the subsequent analysis to maintain the integrity of the matched groups. This stringent methodology, including the use of the PS to obtain a homogeneous cohort, aimed to minimize bias and facilitate a robust assessment of the association between VAP development and patient outcomes, including the additional costs associated with extended hospital stay days.

To quantify the additional expenses associated with prolonged hospital stays, we initially obtained information from the hospital’s economic department regarding the daily expenses per hospital stay in terms of both the CCU and the medical services (MS) during each study season (2012–2019). The obtained data adhere to the regulated rates mandated by the Spanish public health system, which specify that the daily cost per stay include expenses for accommodation, meals, medical assistance, nursing care, infrastructure, public services, technology, supplies, and utility costs such as electricity and water consumption. Following this, the established daily cost per stay was multiplied by the additional days of stay attributable to patients with ventilator-associated pneumonia (VAP). The overall stay costs, CCU stay costs, and medical services (MS) stay costs were computed for each patient, categorized by sex, age group, CVC-BSI, exitus, APACHE II score, and study season. Statistical differences in the mean length of stay between exposed and non-exposed groups were assessed using the Student T-test, with a statistical significance level set at *p* < 0.05.

## 3. Results

### 3.1. Global Population

From 2012 to 2019, the number of patients admitted to critical care units (CCU) who underwent invasive mechanical ventilation (MV) for a period exceeding 48 h was 2720 (Figure 1). The study included patients with complete data for all variables (1976): 449 patients with ventilator-associated pneumonia (VAP) and 1527 without VAP. A comparison was conducted between the patients included in the study (1976) and the patients excluded from the study (744), which revealed no significant differences in terms of sex, age, and the presence of central venous catheter-associated bloodstream infection (CVC-BSI) (Appendix A).

### 3.2. Association Analysis: Calculation and Patient Selection Based on Propensity Score (PS)

Table 1 presents the risk factors associated with VAP. The results of bivariate logistic regression show statistical significance for sex, chronic obstructive pulmonary disease (COPD), neoplasia, renal failure, history of tobacco, history of trauma, immunodeficiency, and Acute Physiology and Chronic Health Evaluation II (APACHE) score. These variables were included in the multivariate logistic regression analysis model, with variables that did not reach statistical significance being excluded. After the multivariate analysis, all the variables that were significant were used to calculate the PS for each patient. Due to its clinical relevance and despite not reaching statistical significance, the exitus variable was also included in the multiple regression model and in the PS calculation.

Table 2 displays the characteristics of patients who developed VAP before and after associating them based on the PS. A total of 434 (96.6%) patients with VAP were grouped according to the calculated PS; for each VAP patient, a non-VAP patient with the same PS (85.5%) or with a difference of less than 0.03 (14.5%) was selected. Fifteen (3.4%) VAP patients were excluded, as no suitable match was found (Figure 1). It is noteworthy that before associating the cases based on the PS, the distribution between patients with VAP and without VAP was not homogeneous, with significant differences observed in some of the study variables. However, after the association based on the PS, the distribution of all variables, except for renal failure, followed a homogeneous distribution.

In Figure 2, we depict the duration of mechanical ventilation in the CCU. Patients diagnosed with VAP exhibited a prolonged mechanical ventilation duration (17.40 ± 8.93 days) in contrast to those without VAP (8.93 ± 7.35), demonstrating statistically significant differences (*p* < 0.001). The overall mean duration of mechanical ventilation across all cases was 13.17 ± 9.20 days.

### 3.3. Cost Analysis

Our cost analysis specifically targeted the additional hospital stay, disaggregating it into two primary components: extra days in the critical care unit (CCU) and supplementary days in medical services (MS). Within our cohort, patients diagnosed with ventilator-associated pneumonia (VAP) exhibited a significantly extended stay in the CCU compared to their counterparts without VAP, with respective durations of 26.20 ± 18.77 days and 12.78 ± 12.33 days (*p* < 0.001). In contrast, the duration of the stay in MS showed no significant differences between the VAP and non-VAP groups: 13.20 ± 18.73 days and 13.06 ± 27.39 days (*p* = 5.04), respectively. When examining specific subgroups, including sex, age, central venous catheter bloodstream infection (CVC-BSI), APACHE II score, and the years studied (2012–2019), the duration of CCU stay remained notably higher in VAP cases (*p* < 0.001). Remarkably, there were no significant variations in the duration of MS stay based on the presence or absence of VAP, as evidenced in Table 3. This influence of VAP on prolonged CCU stay persisted across different levels of patient severity, as indicated by the APACHE II score, with an average increase of 15 days for those with VAP. Additionally, this trend of extended CCU stays in VAP cases compared to MS stays was consistent across all seasons studied from 2012 to 2019 (Table 3).

In the analysis of costs associated with the extended hospital stay due to the development of VAP, the average additional hospital stay was 13.56 days (13.42 extra days in the CCU and 0.14 days in the MS). Considering these extra days and the average cost per day of the stay in the CCU (EUR 1557.10) and in MS (EUR 492.87), the incremental cost associated with each case of VAP was EUR 20,965.28. This resulted in a total additional cost of EUR 12,348,549.92 for all VAP cases (*n* = 589) included in the study period from 2012 to 2019.

## 4. Discussion

Utilizing a methodological approach aimed at achieving cohort homogeneity to ensure that the observed increase in both additional hospital stay and associated costs is attributed specifically to the onset of ventilator-associated pneumonia (VAP) rather than other patient characteristics, our study demonstrates an extended hospital stay of 13.56 days and an additional cost of EUR 20,965.28 per VAP case.

Our findings are similar to those reported in other studies conducted in Europe and the United States, reflecting an additional hospital stay duration of approximately 12 days and an associated extra cost per episode ranging from USD 11,000 to 80,000 [18,19,28,29,30,31,32]. Discrepancies in cost estimates can be attributed to variations in methodological approaches across these studies, resulting in figures that are up to four times higher than those observed in the present study. For instance, in a European study by Leistner et al. (2013), the average hospital cost was significantly higher in patients with lower respiratory tract infection compared to those without infection (EUR 45,401 vs. EUR 26,467; *p* < 0.001), with attributable costs per patient of EUR 17,015, equivalent to USD 23,651 [17]—similar figures to those obtained in this study (EUR 47,301.90 vs. EUR 26,336.62, respectively). Although recent European studies for direct comparison are lacking, a study in Chile by Véliz and Fica et al. (2017) reported a cost exceeding USD 4000 per VAP event, with an overall healthcare cost for VAP surpassing USD 80,000 [32]. Similarly, in Japan, Nanao et al. (2021) disclosed an average additional medical cost of USD 34,884 per VAP episode [30]. This consistency across diverse regions further substantiates the generalizability and robustness of our findings within the broader international context.

In line with several studies, the duration of hospitalization has been identified as the primary factor contributing to increased hospital costs. Several investigations indicate a noteworthy extension in the length of stay within critical care units (CCUs) for patients experiencing VAP, with an approximate increase of 20 days compared to an interval ranging from 6 to 15 days for control subjects [19]. Patients with VAP displayed an average stay in the critical care unit (CCU) of 26.20 ± 18.77 days, in contrast to 12.78 ± 12.33 days for those without infection. These findings align with previously published studies, reflecting durations ranging from 15 to 28 days [18,19,28,29,30,31,32,33].

The average daily cost of CCU stay (EUR 1557.10) closely aligns with the findings reported by Cocanour et al. (2005) (USD 1861 per day). However, in their investigation, each episode of VAP incurred a cost of USD 57,000 [17], whereas in our CCU, the attributable cost was EUR 20,965.28. In a study by Véliz and Fica (2017), a one-day difference was observed in their study cohort for hospitalization outside the CCU. Consistent with our findings of 0.14 days of stay in medical services (MS), this minimal difference underscores that the attributable costs of patients with VAP are primarily a consequence of the prolonged duration of CCU stay (EUR 20,896.28 vs. EUR 69.00) [32].

The length of stay in the CCU was significantly longer in patients with VAP based on sex, age, CVC-BSI, exitus, and APACHE II score (*p* < 0.001). There were no significant differences in the duration of stay in medical services (MS) regardless of the acquisition of VAP (Table 3).

A prolonged hospitalization duration is noted in patients who concurrently manifest both CVC-BSI and VAP, amounting to 56.45 days, in contrast to individuals exclusively experiencing VAP, with a mean hospital stay of 37.13 days. The observed difference is primarily attributable to the prolonged length of stay in the CCU. The elongation of hospital stay is likely a consequence of the clinical complications arising from the simultaneous development of both HAIs in these patients (Table 3). Similarly, a prolonged overall hospital stay, particularly within the CCU, was observed in patients with an APACHE II score exceeding 25. This score is linked to a mortality rate surpassing 50%, elucidating the extended CCU stay for patients with higher severity (Table 3). During the study period, from 2012 to 2019, the duration of hospital stay and, consequently, the attributable extra costs increased significantly, primarily due to CCU stay and its associated costs (Table 3). The detailed breakdown of subgroups such as sex, age, central venous catheter bloodstream infection (CVC-BSI), APACHE II score, and study period (2012–2019) provides a comprehensive understanding of the factors influencing the observed differences in hospital stay duration between VAP and non-VAP groups.

In our study, we conducted sensitivity analyses on two specific subgroups to evaluate potential variations in the economic impact across different patient populations (Table 3). Patients facing both central venous catheter-associated bloodstream infections (CVC-BI) and Ventilator-Associated Pneumonia (VAP) exhibited an extended hospitalization of 56.45 days, contrasting with those exclusively dealing with VAP (mean hospital stay: 37.13 days). The significant difference primarily arose from prolonged stays in the critical care unit (CCU), underscoring complications associated with the simultaneous development of both healthcare-associated infections (HAIs). The second analysis focused on age, aiming to explore the impact of age variations on economic outcomes. Patients aged 65 years and above who developed VAP experienced an extended hospitalization of 37.22 days, in contrast to those without VAP (mean hospital stay: 22.98 days). In both subgroup analyses, prolonged hospital stay emerged as a pivotal contributor to elevated additional costs, predominantly attributed to extended stays in the CCU, as illustrated in Table 3. These sensitivity analyses contribute to a comprehensive understanding of how diverse patient characteristics shape the economic impact of ventilator-associated pneumonia (VAP) in our study.

The extended hospitalization associated with the onset of VAP in the CCU is primarily linked to the extended period patients spend on mechanical ventilation (MV). Patients with VAP exhibited an average duration of MV of 17.40 ± 8.93 days, in contrast to 8.93 ± 7.35 days in those who did not develop VAP (Figure 2). Across the analyzed studies, the mean duration of MV in patients with VAP ranged from 12 to 21 days, compared to from 2 to 10 days in patients who did not develop VAP [18,19,28,29,30,31,32].

In Spain, no studies of this nature, specifically analyzing the increase in the length of hospital stay and the additional cost related to VAP or providing information on costs related to HAIs, have been identified. Published studies reveal significant variations in estimating the overall economic burden and for different types of HAIs analyzed. This variability could be attributed to the heterogeneity in the case definition of VAP used, the diversity of patient populations studied, variations in study designs and methodologies, and, most importantly, the different healthcare systems, each characterized by its own management/financing models and allocation of healthcare resources and expenses. Although this limitation is explicitly acknowledged in the context of healthcare systems cost studies, a shared common factor emerges from the scientific literature evaluated in this study and our results—that is, the notable disparities in expenses between patients with infections and their non-infected counterparts.

Likewise, there have been relatively few studies with the methodology used in this study, in which the propensity score (PS) is employed to obtain a cohort of patients as homogenous as possible, thus limiting the potential influence of other comorbidities that could contribute to increased length of hospital stay. This methodology allows us to assume that VAP plays the primary role in the increased length of hospital stay and associated costs. It is noteworthy that prior to applying the propensity score (PS), there existed a lack of homogeneity between the two study groups, with significant differences identified among them. However, following the application of the PS, no significant differences were found in terms of the epidemiological characteristics and comorbidities of the patients, resulting in a homogeneous distribution (Table 2). This methodology enables us to assert that the development of VAP plays a primary role in the increased length of hospital stay and associated costs. Despite the aforenoted added value, the limited number of studies employing the same methodology makes it challenging to make direct comparisons with other studies. Kollef et al. (2012), in their economic evaluation of the impact of VAP, also used the PS for patient grouping to adjust for heterogeneity between groups. In their study, the additional cost of hospitalization was USD 39,828 [28].

This study is subject to several limitations. Due to incomplete information, the inclusion of all patients from the initial cohort proved challenging, leading to the selection of only those with complete data (Figure 1). Similarly, a subset of patients with VAP had to be excluded from the final study cohort, resulting in the exclusion of 15 patients. This circumstance introduces a potential risk of selection bias. Nonetheless, a thorough comparison between the included and excluded patients revealed no significant differences in terms of sex, age, and BACVC frequency, minimizing the likelihood of selection bias (Appendix A). Notably, a disparity in VAP frequency was observed between the two groups. When comparing exposed and non-exposed cases with the same propensity score (PS), no statistically significant differences were detected, indicating a balanced distribution between the included study groups (Appendix B). Thus, the probability of developing an infection in both groups was practically the same (Table 3).

An additional limitation lies in the potential underestimation of the true total cost associated with VAP development. The expenses provided by the hospital’s economic department in this study encompass only general costs attributed to each service or unit, excluding expenses related to diagnostic tests, diagnostic or therapeutic procedures, and medication consumption. Moreover, there is a likelihood that these results underestimate the current total cost linked to VAP development, considering the potential increase in costs over the years and the possibility of higher costs at present.

Recognizing the potential impact of residual confounding, which signifies the introduction of bias due to unmeasured or inadequately controlled factors associated with both VAP risk and length of stay, is crucial in the context of this study. To address this concern, we employed a comprehensive methodology utilizing the PS to obtain a cohort of patients that was as homogeneous as possible, aiming to minimize the potential impact of residual confounding. This approach was complemented by sensitivity analysis and subgroup analysis. Despite these efforts, it is essential to note that complete elimination of residual confounding may not always be achievable in observational studies. In the context of this study, the complex nature of healthcare data introduces the possibility of unmeasured or unknown factors, including healthcare practices, compliance, etc., influencing both VAP risk and length of stay, along with the associated costs.

The association between ventilator-associated pneumonia (VAP) and the decentralized Spanish healthcare system is influenced by regional autonomy, allowing for the customization of healthcare practices to local needs. Variations in critical care strategies, resource allocation, and patient safety initiatives contribute to the complexity of VAP outcomes across regions. Recognizing these regional differences is crucial for understanding the crucial association between VAP and the Spanish healthcare system.

The intricate connection between VAP and the decentralized Spanish healthcare system is closely tied to regional autonomy, which allows for the adaptation of healthcare practices based on local needs. Variations in critical care strategies, resource allocation, and patient safety initiatives contribute to the complexity of VAP outcomes across regions. Recognizing these regional differences is crucial for comprehending the intricate association between VAP and the Spanish healthcare system. Despite the decentralized nature of our healthcare system and the acknowledged limitations, our results can be broadly generalized, primarily owing to the widespread distribution of the public healthcare system throughout Spain. This system delivers healthcare services to the majority of the population across all autonomous communities and includes tertiary-level hospitals across all country regions with similar characteristics and, consequently, similar costs, thereby establishing a comparable environment. This, along with the employed study design and methodology, amplifies the reliability and applicability of our findings.

In conjunction with these considerations, the increased costs associated with VAP and extended length of stay hold multifaceted clinical significance, impacting both individual patients and the healthcare system. Prolonged hospital stays for VAP patients may signal severe illness or other clinical complications, leading to heightened morbidity and higher utilization of healthcare resources, including personnel, facilities, and medical supplies. Moreover, these added costs apply economic pressure on healthcare systems, impacting budgeting and resource allocation.

The effective implementation of infection surveillance, prevention, and control strategies, with a specific focus on VAP prevention, can result in a decrease in VAP morbidity and mortality rates, along with a reduction in the economic burden associated with prolonged hospital stays and healthcare resource consumption. Evaluating the cost-effectiveness of these strategies is crucial for promptly identifying issues, making well-informed decisions, ensuring optimal patient outcomes, and utilizing resources efficiently.

Recognizing the financial impact emphasizes the importance of preventive measures in reducing the incidence of VAP, thereby underscoring the significance of effective infection control and management practices. Essentially, the potential positive impact of surveillance, prevention, and control programs targeting VAP and healthcare-associated infections (HAIs) may lead to the allocation of resources to expand healthcare services and strengthen preventive initiatives. This, in turn, guides healthcare services towards implementing higher-quality and more cost-effective policies, ultimately advancing the primary goal of fortifying healthcare safety.

## 5. Conclusions

This study allowed for the estimation of the cost per case of VAP in the public healthcare system of Spain, underscoring the significant economic impact of this healthcare issue. Based on the findings derived through the employed methodology and current clinical evidence, it can be concluded that hospital care for patients who develop VAP approximately doubles healthcare expenses compared to patients who do not acquire this type of infection during hospitalization. This cost escalation is further accentuated by evident increases in expenses related to prolonged hospital stays both on a global scale and when analyzed independently for each patient subgroup. The implementation of infection surveillance, prevention, and control strategies, with a specific emphasis on preventing VAP, emerges as a crucial avenue for reducing VAP morbidity and mortality rates, as well as alleviating the economic burden associated with extended hospital stays and healthcare resource consumption, thereby guiding healthcare services toward the implementation of higher-quality and more cost-effective policies and advancing the primary goal of fortifying healthcare safety.

## Figures and Tables

**Figure 1 antibiotics-13-00002-f001:**
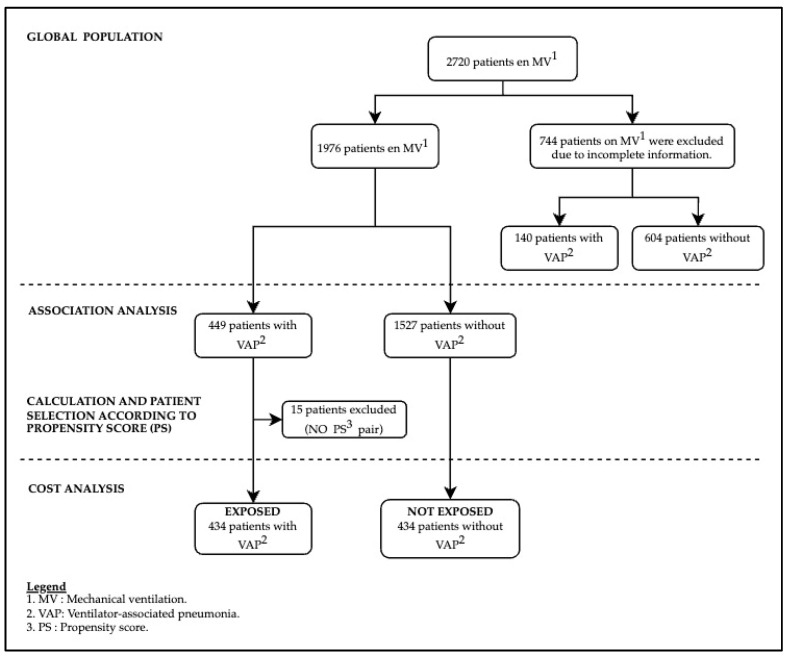
Flowchart of patient selection and study analysis.

**Figure 2 antibiotics-13-00002-f002:**
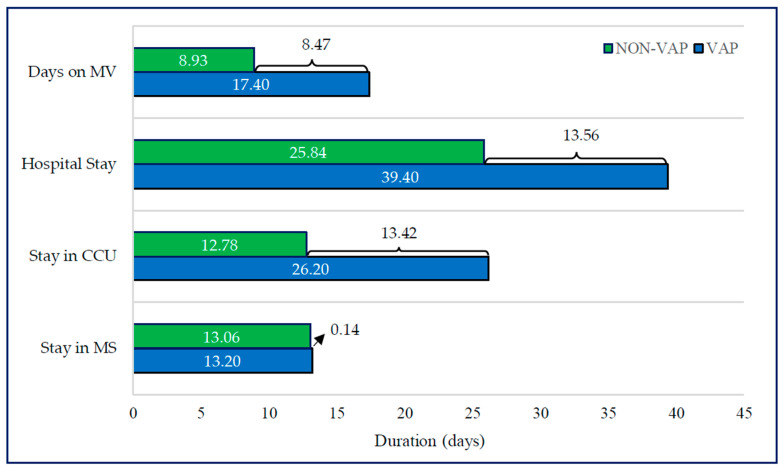
Duration of mechanical ventilation, hospital stay, intensive care unit stay, and medical services stay in the study cohort.

**Table 1 antibiotics-13-00002-t001:** Risk factors associated with ventilator-associated pneumonia (*n* = 1976).

	VAP ^a^ % (*n*)	OR ^b^ (IC 95%)	P ^c^	aOR ^d^ (IC 95%)	aP ^e^
Sex					
Male	25.3 (324/1282)	1.4 (1.17–1.69)	<0.001	1.4 (1.12–1.81)	0.004
Female	18.0 (125/694)	1		1	
Age					
≥ 65 years	22.8 (188/823)	1.0 (0.86–1.19)	0.914	-	-
< 65 years	22.6 (261/1153)	1			
Diabetes mellitus					
Yes	23.7 (97/409)	1.1 (0.87–1.29)	0.590	-	-
No	22.5 (352/1567)	1			
COPD ^f^					
Yes	17.3 (35/202)	0.7 (0.54–1.02)	0.053	0.7 (0.45–1.0)	0.048
No	23.3 (414/1774)	1		1	
Obesity					
Yes	23.1 (37/160)	1.0 (0.76–1.37)	0.899	-	-
No	22.7 (412/1816)	1			
Neoplasia					
Yes	15.8 (44/279)	0.7 (0.50–0.88)	0.003	0.7 (0.46–0.96)	0.027
No	23.9 (405/1697)	1		1	
Neutropenia					
Yes	19.4 (6/31)	0.9 (0.41–1.75)	0.652	-	-
No	22.8 (443/1945)	1			
ARDS ^g^					
Yes	35.3 (12/34)	1.6 (0.99–2.49)	0.078	-	-
No	22.5 (437/1942)	1			
Renal failure					
Yes	18.7 (32/171)	0.8 (0.59–1.12)	0.191	-	-
No	23.1 (417/1805)	1			
MOF ^h^					
Yes	18.8 (9/48)	0.8 (0.45–1.49)	0.506	-	-
No	22.8 (440/1928)	1			
History of tobacco use					
Yes	25.8 (165/639)	1.2 (1.03–1.44)	0.023	1.3 (1.01–1.63)	0.038
No	21.2 (284/1337)	1		1	
History of alcohol use					
Yes	25.6 (60/234)	1.1 (0.91–1.45)	0.257	-	-
No	22.3 (389/1742)	1			
History of trauma					
Yes	33.8 (99/293)	1.6 (1.35–1.96)	<0.001	1.7 (1.25–2.20)	<0.001
No	20.8 (350/1683)	1		1	
Immunodeficiency					
Yes	14.6 (20/137)	0.6 (0.41–0.95)	0.019	0.7 (0.41–1.16)	0.161
No	23.3 (429/1839)	1		1	
ACS ^i^					
Yes	25.7 (86/335)	1.2 (0.95–1.42)	0.158	-	-
No	22.1 (363/1641)	1			
CVC-BSI ^j^					
Yes	44.9 (66/147)	2.1 (1.76–2.62)	<0.001	3.0 (2.08–4.23)	<0.001
No	20.9 (383/1829)	1		1	
Exitus					
Yes	20.8 (155/745)	0.9 (0.73–1.04)	0.114	0.8 (0.66–1.07)	0.151
No	23.9 (294/1231)	1		1	
APACHE ^k^ II score					
0–4	16.0 (24/150)	1		1	
5–9	22.4 (57/254)	1.5 (0.90–2.57)	0.120	1.5 (0.86–2.52)	0.161
10–14	20.9 (76/363)	1.4 (0.84–2.30)	0.201	1.3 (0.78–2.21)	0.301
15–19	26.2 (116/443)	1.9 (1.15–3.03)	0.012	1.9 (1.14–3.12)	0.014
20–24	26.8 (89/332)	1.9 (1.17–3.17)	0.010	2.0 (1.21–3.45)	0.007
25–29	24.7 (59/239)	1.7 (1.02–2.91)	0.043	1.8 (1.03–3.09)	0.039
>30	14.4 (28/195)	0.9 (0.49–1.59)	0.673	1.1 (0.57–1.97)	0.849

^a^ Ventilator-associated pneumonia; ^b^ OR: crude odds ratio; ^c^ P: *p*-value; ^d^ aOR: adjusted odds ratio for sex, COPD ^f^, neoplasia, history of tobacco use, history of trauma, immunodeficiency, CVC-BSI ^j^, exitus, and APACHE II score; ^e^ aP: adjusted *p*-value; ^f^ chronic obstructive pulmonary disease; ^g^ acute respiratory distress syndrome; ^h^ multiorgan failure; ^i^ acute coronary syndrome; ^j^ central venous catheter-associated bloodstream infection (CVC-BSI); ^k^ Acute Physiology and Chronic Health Evaluation II score.

**Table 2 antibiotics-13-00002-t002:** Characteristics of patients who developed ventilator-associated pneumonia before and after propensity score association.

	Before			After		
	VAP ^a^ (*n* = 449) % (*n*)	Non-VAP ^a^ (*n* = 1527) % (*n*)	P ^b^	VAP ^a^ (*n* = 434) % (*n*)	Non-VAP ^a^ (*n* = 434) % (*n*)	P ^b^
Sex			<0.001			0.822
Male	72.2 (324)	62.7 (958)		71.7 (311)	71.0 (308)	
Female	27.8 (125)	37.3 (569)		28.3 (123)	29.0 (126)	
Age			0.914			0.065
≥ 65 years	41.9 (188)	41.6 (635)		41.7 (181)	47.9 (208)	
< 65 years	58.1 (261)	58.4 (892)		58.3 (253)	52.1 (226)	
Diabetes mellitus			0.590			0.774
Yes	21.6 (97)	20.4 (312)		21.9 (95)	22.8 (99)	
No	78.4 (352)	79.6 (1215)		78.1 (339)	77.2 (335)	
COPD ^c^			0.053			0.798
Yes	7.8 (35)	10.9 (167)		7.8 (34)	7.4 (32)	
No	92.2 (414)	89.1 (1360)		92.2 (400)	92.6 (402)	
Obesity			0.899			0.627
Yes	8.2 (37)	8.1 (123)		8.1 (35)	9.0 (39)	
No	91.8 (412)	91.9 (1404)		91.9 (399)	91.0 (395)	
Neoplasia			0.003			0.818
Yes	9.8 (44)	15.4 (235)		9.9 (43)	9.4 (41)	
No	90.2 (405)	84.6 (1292)		90.1 (391)	90.6 (393)	
Neutropenia			0.652			0.762
Yes	1.3 (6)	1.6 (25)		1.4 (6)	1.2 (5)	
No	98.7 (443)	98.4 (1502)		98.6 (428)	98.8 (429)	
ARDS ^d^			0.078			0.508
Yes	2.7 (12)	1.4 (22)		2.8 (12)	2.1 (9)	
No	97.3 (437)	98.6 (1505)		97.2 (422)	97.9 (425)	
Renal failure			0.191			0.033
Yes	7.1 (32)	9.1 (139)		6.9 (30)	11.1 (48)	
No	92.9 (417)	90.9 (1388)		93.1 (404)	88.9 (386)	
MOF ^e^			0.506			1.0
Yes	2.0 (9)	2.6 (39)		2.1 (9)	2.1 (9)	
No	98.0(440)	97.4 (1488)		97.9 (425)	97.7 (425)	
History of tobacco use			0.023			0.724
Yes	36.7 (165)	31.0 (474)		36.6 (159)	35.5 (154)	
No	63.3 (284)	69.0 (1053)		63.4 (275)	64.5 (280)	
History of alcohol use			0.257			0.688
Yes	13.4 (60)	11.4 (174)		13.6 (59)	12.7 (55)	
No	86.6 (389)	88.6 (1353)		86.4 (375)	87.3 (379)	
History of trauma			<0.001			0.869
Yes	22.0 (99)	12.7 (194)		21.2 (92)	21.7 (94)	
No	78.0 (350)	87.3 (1333)		78.8 (342)	78.3 (340)	
Immunodeficiency			0.019			0.496
Yes	4.5 (20)	7.7 (117)		4.6 (20)	3.7 (16)	
No	95.5 (429)	92.3 (1410)		95.4 (414)	96.3 (418)	
ACS ^f^			0.158			0.492
Yes	19.2 (86)	16.3 (249)		18.3 (79)	16.6 (71)	
No	80.8 (363)	83.7 (1278)		81.7 (352)	83.4 (358)	
CVC-BSI ^g^			<0.001			0.916
Yes	14.7 (66)	5.3 (81)		11.8 (51)	11.5 (50)	
No	85.3 (383)	94.7 (1446)		88.3 (383)	88.5 (384)	
Exitus			0.114			0.777
Yes	34.5 (155)	38.6 (590)		35.0 (152)	35.8 (156)	
No	65.5 (294)	61.4 (937)		65.0 (282)	64.1 (278)	
APACHE ^h^ II score			0.004			0.959
0–4	5.3 (24)	8.3 (126)		5.5 (24)	5.5 (24)	
5–9	12.7 (57)	12.9 (197)		12.7 (55)	11.1 (48)	
10–14	16.9 (76)	18.8 (287)		17.1 (74)	18.0 (78)	
15–19	25.8 (116)	21.4 (327)		25.3 (110)	23.3 (101)	
20–24	19.8 (89)	15.9 (243)		20.0 (87)	21.2 (92)	
25–29	13.1 (59)	11.8 (180)		13.1 (57)	14.7 (64)	
>30	6.2 (28)	10.9 (167)		6.2 (27)	6.2 (27)	

^a^ Ventilator-associated pneumonia; ^b^ P: *p*-value; ^c^ chronic obstructive pulmonary disease; ^d^ acute respiratory distress syndrome; ^e^ multiorgan failure; ^f^ acute coronary syndrome; ^g^ central venous catheter-associated bloodstream infection; ^h^ Acute Physiology and Chronic Health Evaluation II score.

**Table 3 antibiotics-13-00002-t003:** Analysis of costs associated with increased hospital stay related to the development of ventilator-associated pneumonia (*n* = 868).

		Critical Care Units (CCUs ^a^)						Medical Services (MS ^b^)						**Cost Evaluation**	
	Total Stay Dif ^c^	CCU ^a^ Stay Exposed	CCU ^a^ Stay Non-Exposed	*p*-Value	CCU ^a^ Stay Dif ^c^	CPDS ^f^ in CCU ^a^	Extra CCU ^a^ Cost	MS ^b^ Stay Exposed	MS ^b^ Stay Non-Exposed	*p*-Value	MS ^b^ Stay Dif ^c^	CPDS ^f^ in MS ^b^	**Extra MS ^b^ Cost**	**Overall CPDS ^f^**	**Total Extra Cost**
	(Days)	(µ ^d^ ± σ ^e^)	(µ ^d^ ± σ ^e^)		(Days)	(EUR)	(EUR)	(µ ^d^ ± σ ^e^)	(µ ^d^ ± σ ^e^)		(Days)	(EUR)	**(EUR)**	**(EUR)**	**(EUR)**
TOTAL (N = 868)	13.56	26.20 ± 18.77	12.78 ± 12.33	<0.001	13.42	1557.10	20.896.28	13.20 ±18.73	13.06 ± 27.39	0.934	0.14	492.87	**69.00**	**2049.97**	**20,965.28**
Sex															
Male (N = 619)	14.99	26.50± 18.64	12.69± 10.97	<0.001	13.81	1557.10	21,503.55	13.39 ± 19.58	12.21 ± 23.30	0.498	1.18	492.87	581.59	2049.97	22,085.14
Female (N = 249)	10.04	25.46 ± 19.17	13.00 ± 15.21	<0.001	12.46	1557.10	19,401.47	12.72 ± 16.47	15.14 ± 35.48	0.491	−2.42	492.87	−1192.75	2049.97	18,208.72
Age															
≥65 (N = 389)	14.94	26.70 ± 20.53	11.09 ± 11.64	<0.001	15.61	1557.10	24,306.33	10.52 ± 15.84	11.19 ± 25.08	0.758	−0.67	492.87	−330.22	2049.97	23,976.11
<65 (N = 479)	11.84	25.85 ± 17.44	14.33 ± 12.76	<0.001	11.52	1557.10	17,937.79	15.11 ± 20.37	14.79 ± 29.30	0.891	0.32	492.87	157.72	2049.97	18,095.51
CVC-BSI ^g^															
Yes (N = 101)	16.21	39.90 ± 21.65	21.18 ± 16.05	<0.001	18.72	1557.10	29,148.91	16.55 ± 23.29	19.06 ± 24.89	0.602	−2.51	492.87	−1237.10	2049.97	27,911.81
No (N = 767)	13.17	24.38 ± 17.60	11.68 ± 11.34	<0.001	12.70	1557.10	19,775.17	12.75 ± 18.03	12.28 ± 27.63	0.782	0.47	492.87	231.65	2049.97	20,006.82
Exitus															
Yes (N = 308)	10.62	24.84 ± 21.45	9.74 ± 11.24	<0.001	15.10	1557.10	23,512.21	3.44 ± 9.26	7.92 ± 39.69	0.176	−4.48	492.87	−2208.06	2049.97	21,304.15
No (N = 560)	14.95	26.93 ± 17.16	14.48 ± 12.61	<0.001	12.45	1557.10	19,385.90	18.45 ± 20.38	15.95 ± 16.36	0.110	2.50	492.87	1232.18	2049.97	20,618.08
APACHE ^h^ II score															
0–4 (N = 48)	13.24	27.58 ± 21.99	13.63 ± 13.60	0.011	13.95	1557.10	21,721.55	12.08 ± 15.71	12.79 ± 13.73	0.869	−0.71	492.87	−349.94	2049.97	21,371.61
5–9 (N = 103)	19.41	29.76 ± 19.47	11.42 ± 11.92	<0.001	18.34	1557.10	28,557.21	14.53 ± 17.69	13.46 ± 14.47	0.740	1.07	492.87	527.37	2049.97	29,084.58
10–14 (N = 152)	18.41	26.08 ± 21.99	12.27 ± 13.31	<0.001	13.81	1557.10	21,503.55	16.61 ± 23.16	12.01 ± 17.21	0.166	4.60	492.87	2267.20	2049.97	23,770.75
15–19 (N = 211)	0.96	25.09 ± 16.73	14.65 ± 12.44	<0.001	10.44	1557.10	16,256.12	10.28 ± 12.81	19.76 ± 49.74	0.055	−9.48	492.87	−4672.41	2049.97	11,583.71
20–24 (N = 179)	11.22	22.91 ± 14.81	13.77 ± 13.34	<0.001	9.14	1557.10	14,231.89	11.16 ± 14.34	9.08 ± 13.19	0.312	2.08	492.87	1025.17	2049.97	15,257.06
25–29 (N = 121)	21.78	28.12 ± 19.96	9.78 ± 7.84	<0.001	18.34	1557.10	28,557.21	15.16 ± 26.16	11.72 ± 14.82	0.369	3.44	492.87	1695.47	2049.97	30,252.68
>30 (N = 54)	25.56	29.11 ± 21.24	12.59 ± 13.31	0.001	16.52	1557.10	25,723.29	16.41 ± 22.86	7.37 ± 11.80	0.074	9.04	492.87	4455.54	2049.97	30,178.83
Periods															
2012 (N = 64)	−6.59	24.20 ± 15.73	14.85 ± 11.94	0.021	9.35	1266.51	11,841.87	15.11 ± 20.86	31.05 ± 80.00	0.219	−15.94	430.55	−6862.97	1697.06	4978.90
2013 (N = 86)	16.98	21.66 ± 12.55	11.80 ± 10.66	<0.001	9.86	1615.54	15,929.22	10.34 ± 18.25	6.10 ± 9.59	0.239	4.24	503.38	2234.33	2118.92	18,163.55
2014 (N = 91)	13.52	28.95 ± 19.62	14.48 ± 10.88	<0.001	14.47	1585.23	22,938.28	13.22 ± 16.73	14.17 ± 19.93	0.812	−0.95	467.30	−443.94	2052.53	22,494.34
2015 (N = 128)	11.76	30.03 ± 25.25	14.98 ± 15.28	<0.001	15.05	1491.00	22,439.55	15.05 ± 24.17	18.34 ± 40.82	0.581	−3.29	457.42	−1504.91	1948.42	20,934.64
2016 (N = 127)	18.84	29.35 ± 20.04	16.08 ± 15.03	<0.001	13.27	1451.67	19,263.66	14.18 ± 18.26	8.61 ± 10.45	0.041	5.57	480.58	2676.83	1932.25	21,940.49
2017 (N = 154)	12.32	23.05 ± 14.33	11.30 ± 10.03	<0.001	11.75	1654.55	19,440.96	12.10 ± 14.56	11.53 ± 15.62	0.822	0.57	493.16	281.10	2147.70	19,722.06
2018 (N = 117)	15.89	25.69 ± 18.14	11.02 ± 12.32	<0.001	14.67	1579.05	23,164.66	12.48 ± 17.16	11.26 ± 18.83	0.718	1.22	518.59	632.68	2097.64	23,797.34
2019 (N = 101)	18.21	26.33 ± 19.73	8.84 ± 9.42	<0.001	17.49	1716.40	30,019.84	13.13 ± 18.45	12.41 ± 14.54	0.827	0.72	583.32	419.99	2299.71	30,439.83

^a^ CCU: critical care units; ^b^ MS: medical services; ^c^ difference; ^d^ µ: mean; ^e^ σ: standard deviation; ^f^ cost per day of stay; ^g^ central venous catheter-associated bloodstream infection; ^h^ Acute Physiology and Chronic Health Evaluation.

## Data Availability

The datasets used and/or analyzed during the current study are available from the corresponding author on reasonable request.

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
