# Peer review of "Estimation of Additional Costs in Patients with Ventilator-Associated Pneumonia"

_antibiotics, 2023, doi:10.3390/antibiotics13010002_

Round 1

Reviewer 1 Report

Comments and Suggestions for Authors

The authors have made a good attempt to estimate additional costs as a result of extended hospital stay in patients with ventilator-associated pneumonia in a large tertiary hospital. A Propensity Score association approach was employed, and the study spans over 5 years with enough participants. However, few issues need to be addressed or improved.

1.     Line 139: How does the overall mean duration of MV (13.17 days) differ from the duration of MV (17.4 days)? Additional explanation should be added to the text.

2.     Line 155: The abbreviation, ‘Apache’, should be written in uppercase throughout the manuscript and written in full at first usage.

3.     Line 215: “A prolonged hospitalization duration is noted in patients who concurrently manifest both CVC-BS and VAP, amounting to 56.45 days, in contrast to individuals exclusively experiencing VAP with a mean hospital stay of 37.13 days.”

These data are not apparent in the results section. Are they contained in the Annex I document? Besides, no such document was provided. Please clarify.

4.     Line 335: “Variables that remained statistically significant in the multivariate model were used to calculate the Propensity Score (PS) for each patient.”

Additional details should be provided on how the Propensity Score was calculated for each patient.

Comments on the Quality of English Language

English Language is satisfactory.

Reviewer 2 Report

Comments and Suggestions for Authors

Thank you for the opportunity to review this manuscript. The manuscript attempts to raise the issue of additional costs caused by ventilator-associated pneumonia. For this manuscript to be publishable, I suggest several major revisions and minor revisions.

MAJOR REVISIONS

1. Materials and methods cannot be presented after results and discussion. This is confusing. Usually there should be an introduction, materials and methods, results, discussion, then conclusion. Please follow this order.

2. It is not clear which study design was used. In the abstract on line 23 you stated, ‘This prospective cohort study’, yet on line 364 you stated ‘A retrospective cohort study…’ Please explain this discrepancy.

3. Materials and Methods need to be improved. You need to explain how propensity scores were calculated in this section, not under results. Furthermore, since a cost analysis was performed, you need to state the assumptions in the calculations.

4. Usually sensitivity analyses are performed when doing economic evaluations. Is there a reason why this was not done?

5. The manuscript should have a stand-alone conclusion section.

MINOR REVISIONS

6. Where you state the author’s name in-text, please put the year of publication in brackets. This was not done in lines 62, 64, 202, 204, and 257.

7. In line 80, you state, ‘which is considered one of the best healthcare systems worldwide.’ Please provide a reference to this statement.

8. In line 86, you state, ‘Such researche….’ Please replace word ‘researche’ with ‘research’.

9. In lines 132 and 134, you state ‘PS score’ yet PS represents propensity score. Rather say PS value.

10. In line 178, you state, ‘….reflecting and additional hospital stay….’ Replace ‘and’ with ‘an’.

11. In line 326, you state, ‘For the association study…..’ This is not clear. Was there another study separate from this one, or you mean association analysis?

Comments on the Quality of English Language

Minor editing required

Reviewer 3 Report

Comments and Suggestions for Authors

While this study addresses the important issue of excess healthcare costs attributable to ventilator-associated pneumonia (VAP), several aspects of the methodology and presentation of findings limit the validity and generalizability of the results.

Major concerns:

Methods

The methods lack details on key aspects of the analysis:

·         Selection bias:

o   The significantly lower rate of VAP among excluded patients suggests exclusion criteria or missing data elements may be related to VAP status. Implications of this potential selection bias are not discussed.

·         Propensity score modeling:

o   The rationale and process for selecting variables to include in the propensity score model is not provided. Some justification should be given for which factors are potential confounders.

o   There is no mention of checking the balance or overlap of the propensity score distributions between VAP and non-VAP groups. This is an important validation step.

·         Confounding factors:

o   Residual confounding could still exist if key factors that influenced both VAP risk and length of stay were not properly accounted for in the propensity score model and matching process. This issue is not addressed.

·         Cost analysis:

o   The source of the cost per day estimates is not described. Are these based purely on accounting data? If so, they likely underestimate true economic costs. Was inflation considered?

o   No sensitivity analyses were done to evaluate the effect of varying costs.

·         Statistical analysis:

o   The methods state statistical tests were used to compare lengths of stay between groups, but no details are provided on the specific tests used.

Results

·         It is unclear from the text if the extra costs around €21,000 per VAP case is considered high or low compared to other settings.

·         Clinical significance of the length of stay and cost differences is not discussed later

·         Data tables are extensive, but the key takeaways are not highlighted

Discussion

·         Findings are compared superficially to previous studies without depth

·         Contextualization within Spanish healthcare system and policy implications are not provided

·         Limitations around residual confounding, cost estimates, generalizability deserved more attention

·         The claim about doubling/tripling costs due to VAP is not firmly supported by the data

Round 2

Reviewer 2 Report

Comments and Suggestions for Authors

The authors have addressed all my comments. 

Reviewer 3 Report

Comments and Suggestions for Authors

Thanks to the authors for addressing my previous comments. I don't have further comments.